# Tetrabromobisphenol Exposure Impairs Bovine Oocyte Maturation by Inducing Mitochondrial Dysfunction

**DOI:** 10.3390/molecules27228111

**Published:** 2022-11-21

**Authors:** Jing Guo, Chang-Guo Min, Kai-Yan Zhang, Cheng-Lin Zhan, Yu-Chan Wang, Sheng-Kui Hou, Xin Ma, Wen-Fa Lu

**Affiliations:** 1Key Lab of Animal Production, Product Quality and Security, Ministry of Education, Jilin Agricultural University, Changchun 130118, China; 2College of Animal Science and Technology, Jilin Agricultural University, Changchun 130118, China; 3Engineering Research Center of North-East Cold Region Beef Cattle Science & Technology Innovation, Ministry of Education, Yanbian University, Yanji 133002, China

**Keywords:** TBBPA, oocytes, mitochondria, oxidative stress

## Abstract

Tetrabromobisphenol (TBBPA) is the most widely used brominated flame retardant in the world and displays toxicity to humans and animals. However, few studies have focused on its impact on oocyte maturation. Here, TBBPA was added to the culture medium of bovine cumulus–oocyte complexes (COCs) to examine its effect on oocytes. We found that TBBPA exposure displayed an adverse influence on oocyte maturation and subsequent embryonic development. The results of this study showed that TBBPA exposure induced oocyte meiotic failure by disturbing the polar-body extrusion of oocytes and the expansion of cumulus cells. We further found that TBBPA exposure led to defective spindle assembly and chromosome alignment. Meanwhile, TBBPA induced oxidative stress and early apoptosis by mediating the expression of superoxide dismutase 2 (SOD2). TBBPA exposure also caused mitochondrial dysfunction, displaying a decrease in mitochondrial membrane potential, mitochondrial content, mtDNA copy number, and ATP levels, which are regulated by the expression of pyruvate dehydrogenase kinase 3 (PDK3). In addition, the developmental competence of oocytes and the quality of blastocysts were also reduced after TBBPA treatment. These results demonstrated that TBBPA exposure impaired oocyte maturation and developmental competence by disrupting both nuclear and cytoplasmic maturation of the oocyte, which might have been caused by oxidative stress induced by mitochondrial dysfunction.

## 1. Introduction

Tetrabromobisphenol (TBBPA) is the most applied brominated flame retardant in the world, with production reaching over 2 million tons per year, which represents about 60% of global flame retardant production [1]. It is used in a range of paper, textile, electrical, and electronic appliances. It is detectable in ubiquitous environmental media, including air, water, soil, indoor dust, sediments, and sewage sludge. TBBPA can be bioaccumulated in animals and humans via diet, ingestion, and dermal contact with dust, which poses a threat to health [2]. Recent studies reported that TBBPA is present in many products, such as aquatic food, meat, egg, and especially milk [2,3]. Furthermore, TBBPA can be even detected in human maternal serum, breast milk, and umbilical cord serum [4].

Due to its extensive usage, more and more attention is paid to the toxicity of TBBPA. Several studies have found the potential hazards of TBBPA, including neurotoxicity, cytotoxicity, hepatotoxicity, and others. Therefore, TBBPA is listed in group 2A, “Probably carcinogenic to humans”, by International Agency for Research on Cancer, which is based on sufficient evidence from animal experiments on carcinogenicity [5]. In addition, TBBPA has been demonstrated to be an endocrine disrupter that interferes with hormone synthesis, secretion, transport, and elimination to influence metabolism, hormonal homeostasis, and developmental processes [5,6]. The reproductive toxicity of TBBPA has also aroused concern. In the male reproductive system, a number of studies have displayed that TBBPA exposure could induce reproductive organ damage, sperm quality decrease, and hormone level disorders [7]. Further research has shown that the reproductive toxicity of TBBPA is associated with oxidative stress, which leads to the reduction in sperm quantity and quality [8,9,10]. Importantly, several studies have also showed that TBBPA exhibits serious injury to mitochondria and induces the generation of ROS from mitochondria [5,11,12]. However, there are few studies that have paid attention to its toxicity to oocytes. 

Oocyte maturation is a complex process. During oocyte maturation, mitochondria play important roles. Mitochondrial dysfunction results in reactive oxygen species (ROS) generation and ATP deficiency, inducing oxidative stress [13,14]. Although ROS are natural products of oocyte metabolism, supraphysiological levels of ROS may exceed the scavenging capacity to destroy the cellular antioxidant system, which affects oocyte maturation and early embryonic development [15,16,17]. It is believed that impaired mitochondrial function may lead to the failure of oocyte maturation.

As a worldwide flame retardant, TBBPA toxicity has been demonstrated in previous studies [18,19,20]. However, there are few studies on the effect of TBBPA on domestic animal reproduction. In the present study, the effect and mechanism of TBBPA on oocyte maturation and development competence were determined using a bovine model. We examined the impact of TBBPA on bovine oocyte maturation by assessing the critical events of nuclear and cytoplasmic maturation, including spindle assemble, oxidative stress, mitochondrial function, and apoptosis. We also studied oocyte development competence by evaluating fertilization ability and embryonic development. For the first time, we detected the toxicity of TBBPA to female gamete development and the underlying mechanisms regarding how TBBPA affects oocyte maturation.

## 2. Results

### 2.1. TBBPA Exposure Disrupts Bovine Oocyte Maturation and Cumulus-Cell Expansion

To detect the toxicity of TBBPA to bovine oocyte maturation, different concentrations of TBBPA (0, 20, 40, and 80 µM) were used in the culture medium. As shown in Figure 1A, TBBPA exposure impaired polar-body extrusion and cumulus-cell expansion. However, GVBD was not affected after TBBPT treatment (Figure 1B). The quantitative analyses showed that all TBBPA treatment groups decreased the maturation rate to varying degrees compared with the control group, as reported in Figure 1C (ctrl, 87.78 ± 2.89%; 20 µM, 77.66 ± 2.71%; 40 µM, 65.82 ± 3.17%; 80 µM, 59.05 ± 3.13%). Because the 40 µM and 80 µM TBBPA treatments showed dramatic significance, 40 μM TBBPA was used in subsequent experiments. Cumulus-cell expansion was quantified using the CEI. TBBPA treatment decreased the mean CEI significantly (Figure 1D). Confirming this, the mRNA expression level of expansion-related genes, TNFAIP6 and PTX3, also decreased significantly (Figure 1E). The apoptosis of cumulus cells was also examined. As shown in Figure 1F,G, the apoptosis of cumulus cells increased after TBBPA treatment (*p* < 0.01). In addition, apoptosis was further confirmed by the ratio of BAX to BCL-2 (Figure 1H). These results demonstrated that TBBPA exposure disrupted oocyte meiotic progression and cumulus-cell expansion.

### 2.2. TBBPA Exposure Destroys Spindle Morphology and Chromosome Alignment during Bovine Oocyte Maturation

Normal spindle morphology with aligned chromosomes is one of the most important indicators of high-quality oocytes. Therefore, oocyte spindle assembly and chromosome alignment were examined with immunofluorescent staining. As shown in Figure 2A, the oocytes displayed a typical barrel-shaped spindle with an aligned chromosome on the equatorial plate in the control group. By contrast, in the TBBPA-treated group, the proportion of abnormal spindle morphology with misaligned chromosomes was increased significantly (Figure 2B,C). Oocyte maturation is a precision-coordinated process that is controlled by spindle assembly and chromosome alignment. An increased frequency of abnormal spindle and chromosome misalignment indicates that the microtubule stability and attachment between spindle and chromosome may be impaired after TBBPA exposure. These abnormalities can induce fertilization failure and subsequent embryonic development arrest. These results suggested that TBBPA exposure led to spindle defects and chromosome misalignment. 

### 2.3. TBBPA Exposure Causes Oxidative Stress to Induce Early Apoptosis

Previous studies have demonstrated that the toxicity of TBBPA is associated with oxidative stress [21,22,23]. We hypothesized that TBBPA exposure induced excessive ROS generation to result in oxidative stress. To confirm this hypothesis, ROS, GSH, and DHE levels were detected. The immunofluorescence results showed that ROS and DHE levels were significantly increased (Figure 3A,E) and that the GSH level was decreased after TBBPA treatment (Figure 3C). The measurement of fluorescence intensity was consistent with this (Figure 3B,D,F). To verify the effect of TBBPA on oxidative stress, superoxide dismutase 2 (SOD2) was analyzed. As a scavenger of mitochondrial superoxide, the SOD2 protein level was decreased significantly compared with the control group (Figure 3G,H). These results showed that TBBPA induced oxidative stress by reducing ROS scavenging ability.

Next, early apoptosis was detected after TBBPA treatment. As shown in Figure 3I, the annexin-v signal was barely detected in the control group. However, TBBPA-treated oocytes showed an obvious fluorescence signal. The rate of apoptosis was significantly increased in TBBPA-treated oocytes (Figure 3J). These results indicated that TBBPA exposure impaired oocyte maturation because of oxidative-stress-induced apoptosis.

### 2.4. TBBPA Exposure Induces the Mitochondrial Dysfunction

Mitochondria are the main organelles for ROS generation and are regarded to be among the most important markers of cytoplasmic maturation of oocytes. Therefore, mitochondrial function was examined. The mitochondria content of oocytes was first detected. The fluorescence results displayed that TBBPA exposure led to mitochondria signal reduction (Figure 4A). The quantitative analysis also demonstrated that the fluorescence intensity of mitochondria decreased dramatically (Figure 4B). Furthermore, the mtDNA copy number also decreased significantly (Figure 4C). The mitochondrial membrane potential (ΔΨm) is an essential indicator of mitochondrial function. ΔΨm is a prerequisite for mitochondrial oxidative phosphorylation and the production of ATP. At the same time, it is conducive to maintaining the normal physiological functions of cells [24]. Oocytes from different groups were collected to evaluate mitochondrial function. ΔΨm was detected using JC-1 staining in control and TBBPA groups. As shown in Figure 4D, JC-1 monomers displayed an obvious green signal, and JC-1 aggregates were the opposite. ΔΨm revealed a sharp decline after TBBPA treatment (Figure 4E). Altogether, these results suggested that TBBPA exposure impaired mitochondrial function.

### 2.5. TBBPA Exposure Causes Mitochondrial Dysfunction by Regulating PDK Activity

Mitochondria are the primary sites for ATP production, which is necessary to produce energy for cellular biological events. The ATP content was also examined after TBBPA treatment. As presented in Figure 5A, the ATP signal was much weaker in the TBBPA group than that in the control group. The fluorescence intensity was decreased significantly in TBBPA-treated oocytes (Figure 5C). Pyruvate dehydrogenase kinase (PDK) is a key factor to participate in pyruvate metabolism. We speculated that TBBPA disrupted ATP supplements by regulating the PDK pathway. To confirm this, PDK3 was determined after TBBPA exposure. We found that PDK3 fluorescence intensity increased significantly (Figure 5B,D). These results indicated that TBBPA impaired oocyte maturation because of PDK3-induced ATP deficiency.

### 2.6. TBBPA Exposure Reduces Oocyte Competence

To test oocyte competence, oocytes were fertilized and cultured until the blastocyst stage. We found that the majority of oocytes could be fertilized and developed to the two-cell stage in the control group (76.63%), while fewer TBBPA-treated oocytes could develop to the two-cell stage (66.24%). The fertilization ability of TBBPA-exposed oocytes decreased significantly. The embryos were cultured until the blastocyst stage. The results showed that the four-cell rate (71.68% vs. 52.83%), eight-cell rate (62.38% vs. 47.44%), morula stage (53.89% vs. 42.04%), and blastocyst rate (32.37% vs. 15.06%) were also dramatically reduced in the TBBPA-treated-oocyte group (Figure 6A,B). In addition, we also found that the apoptotic rate of blastocysts also decreased significantly after TBBPA treatment (Figure 6C,D). These results displayed that TBBPA exposure induced the poor quality of oocytes, which resulted in the decreased competence of IVF embryos.

## 3. Discussion

Oocyte maturation is a complicated progression, and changes in the internal and external environment give rise to fertilization failure, embryonic development arrest, and even abortion. Increasing evidence demonstrates that environmental and industrial pollution can disrupt oocyte maturation and even cause infertility [25,26,27]. As the most widely used brominated flame retardant, TBBPA has confirmed toxicity, including neurotoxicity, immunotoxicity, cytotoxicity, etc. However, the influences of TBBPA on female germ cells are unknown. The oocyte is the most important cell in female reproduction. Therefore, the effect of TBBPA on oocytes was examined in this study.

For this purpose, oocyte maturation indicators and related signaling pathways were examined after COCs were treated with TBBPA. Our results showed that TBBPA treatment impaired oocyte meiotic progression and the expansion of cumulus cells. Meanwhile, cumulus-cell apoptosis was also induced after TBBPA exposure. The bidirectional communication between the oocyte and cumulus was crucial to producing a healthy oocyte. Cumulus-cell expansion is necessary for oocyte maturation and subsequent embryonic development. Cumulus cells supply substrates for energy metabolism and biosynthesis to the oocyte [28,29]. Previous studies have also showed that cumulus-cell apoptosis is regarded as a marker for oocyte competence [30]. Therefore, TBBPA displays reproductive toxicity not only to oocyte maturation but also to cumulus cells, which in turn regulate oocyte maturation. 

Oocyte maturation contains nuclear and cytoplasm maturation. The effect of TBBPA on the nuclear maturation of oocytes was confirmed with experiments that showed an increased rate of disorganized spindle assembly and misaligned chromosomes. The accurate control of spindle assembly and chromosome alignment are indispensable to producing a good-quality oocyte [31,32]. The abnormality of spindle assembly and chromosome alignment causes aneuploidy and subsequent embryonic development arrest [33]. The ROS level is a key indicator of cytoplasm maturation [34,35]. The excessive accumulation of ROS induces oxidative stress. As expected, TBBPA treatment significantly elevated ROS and DHE levels and decreased GSH levels. GSH is regarded as an antioxidant that eliminates ROS [36]. In addition, SOD2, which is the main scavenger of ROS, was significantly decreased. TBBPA may reduce the ability to clear ROS by regulating SOD2. ROS accumulation caused the occurrence of early apoptosis, which was confirmed with Annexin-V staining. TBBPA caused the excessive accumulation of ROS by decreasing the ability to scavenge ROS, further inducing early apoptosis. These results present that TBBPA exposure reduced oocyte quality by impairing both nuclear and cytoplasmic maturation of oocytes.

Mitochondria are the main organelles for ROS production and the main targets of ROS. Mitochondria also play an important role in programmed cell death, and their morphology is critical for apoptosis [37]. Therefore, mitochondria are essential for ATP synthesis and redox maintenance [38,39]. In addition, mitochondria also supply ATP to organize spindles and arrange chromosomes during the oocyte meiotic process [40]. Mitochondrial dysfunction can lead to oocyte maturation failure [41]. Therefore, the function of mitochondria is essential for oocyte maturation. The function of mitochondria was evaluated by examining the content of mitochondria, mtDNA copy number, and mitochondrial membrane potential. We found that TBBPA exposure resulted in mitochondrial dysfunction. These results suggested that TBBPA impaired mitochondria function, inducing oxidative stress. 

The mechanism of TBBPA and its effect on oocyte maturation is to be further studied. Mitochondria are energy factories that ensure biological processes. It is known that oocytes cannot utilize glucose directly, which needs to be transformed into pyruvate by cumulus cells. Pyruvate is the main energy source throughout oocyte maturation [42]. Pyruvate is converted to acetyl coenzyme A by the PDK-mediated phosphorylation of pyruvate dehydrogenase (PDH) and enters into the tricarboxylic acid cycle to produce ATP. PDK3 shows higher affinity with the pyruvate dehydrogenase complex than other PDKs [43]. In addition, PDK3 mRNA is most abundant than others in mouse oocytes. A previous study illustrated that PDK3, which is considered to be the primary PDK, regulates metabolism via the phosphorylation of PDH. This study also showed that the overexpression of PDK3 not only causes metabolic dysfunction but also induces spindle assembly defects and chromosome misalignment [43]. Therefore, PDK3 displays an effect on spindle organization besides regulation metabolism. Accordingly, our results showed that TBBPA exposure decreased the ATP contents and increased the PDK3 level. These results illustrated that TBBPA exposure destroyed pyruvate usage via the PDK3 axis to disturb ATP generation. In addition, TBBPA-induced meiotic defects may be associated with PDK3. However, the mechanism needs to be further studied. 

## 4. Materials and Methods

All the chemicals and reagents used for this study were purchased from Sigma-Aldrich (St. Louis, MO, USA), unless otherwise stated.

### 4.1. Oocyte Collection and In Vitro Maturation (IVM) 

Bovine ovaries were collected from a local abattoir and transported to the laboratory within 2 h at 38 °C in phosphate-buffered saline (PBS). Bovine cumulus–oocyte complexes (COCs) were aspirated from small antral follicles (2–8 mm in diameter). Oocytes surrounded by intact cumulus layers were washed 5 times in IVM medium composed of TCM199 (11150-59; Gibco, NY, USA), 0.57 mmol/L cysteine, 10% fetal bovine serum (FBS), 10 μg/mL follicle-stimulating hormone, 0.04 mg/mL pyruvate, 1 μg/mL estradiol, 10 ng/mL epidermal growth factor, and 1% penicillin–streptomycin solution. The IVM dishes had 4 wells per plate (10034; SPL Lifesciences, Pocheon, Korea), and all the oocytes were covered with mineral oil in a humidified atmosphere of 5% CO_2_ at 38.5 °C for 22 h.

### 4.2. TBBPA Treatment

The COCs were randomly divided into the following four groups: control, 20 μM TBBPA, 40 μM TBBPA, and 80 μM TBBPA. The control group was cultured in IVM medium, while the other groups were treated with TBBPA (80 mM in DMSO), which was diluted to 20 μM, 40 μM, and 80 μM with IVM medium. The final concentration of DMSO was 0.1%.

### 4.3. Evaluation of Cumulus Expansion 

After 22 h of the culturing of COCs, the cumulus expansion index (CEI) was calculated according to a previous study [44]. Briefly, cumulus expansion could be divided into five levels: grade 0, no cumulus expansion, oocytes attached to the bottom of the dish; grade 1, only the outermost 1–2 cumulus granulosa cells expanded; grade 2, the outer cumulus granulosa cells expanded radially, and the whole COCs were observed to be fluffy; grade 3, the radial crown part did not expand, while the rest expanded; grade 4, all cumulus granulosa cells expanded. CEI = [(number of grade 0 oocytes × 0) + (number ofgrade1ocytes × 1) + (number of grade 2 oocytes × 2) + (number of grade 3 oocytes × 3) + (number of grade 4 oocytes × 4)]/total number of oocytes.

### 4.4. Measurement of Mitochondrial Membrane Potential (ΔΨm), Reactive Oxygen Species (ROS), Glutathione (GSH), Dihydroergotamine (DHE), Mitochondria, ATP, and Apoptosis

After 22 h of the culturing of COCs, cumulus cells were removed using 0.2% hyaluronidase for 3 min at 38.5 °C. Denuded MII-stage oocytes from different groups were separately treated with fluorescent probes. Fluorescent signals were captured under the same conditions. Oocytes were incubated with 2 μM JC-1 (Beyotime, Shanghai, China) for mitochondrial membrane potential measurement (Beyotime, Shanghai, China), with 10 µM DCFH-DA (Beyotime, Shanghai, China) for ROS level measurement, with 10 µM Cell TrackerTM Blue (Invitrogen, California, USA) for GSH level measurement, with 10 µM Dihydroethidium (Beyotime, Shanghai, China) for DHE level measurement, with Mito-Tracker Red CMXRos (Beyotime, Shanghai, China) for mitochondria analysis, and with Annexin-v (Beyotime, Shanghai, China) for apoptosis observation. All reactive dyes were applied for 30 min at 38.5 °C. The fluorescence intensity was observed using a fluorescence microscope (Olympus, Tokyo, Japan). For JC-1 detection, the JC-1 monomeric form was observed at excitation/emission wavelengths of 488/530 nm, and the JC-1 aggregate form was observed at excitation/emission wavelengths of 529/590 nm. The oocytes were monitored at excitation/emission wavelengths of 488/530 nm to observe ROS, ATP, and apoptosis levels. GSH levels were observed at excitation/emission wavelengths of 371/464 nm. DHE level and mitochondria were observed at excitation/emission wavelengths of 529/590 nm. 

### 4.5. Immunofluorescence

Oocytes, cumulus cells, and blastocysts were washed in PBS containing 0.1% polyvinyl alcohol (PBS-PVA), fixed for 30 min in 3.7% formaldehyde, and permeabilized with 0.5% Triton X-100 in PBS-PVA for 30 min at room temperature. Oocytes were then blocked using 1% BSA in PBS. Next, oocytes were incubated with rabbit anti-PDK3 antibody (1:100; Novus Biologicals, MA, USA), α-tubulin-FITC antibody (1:50; Beyotime, Shanghai, China), and SOD2 (1:100; Proteintech, IL, USA) at 4 °C overnight, followed by incubation with Alexa Fluo 488-conjugated secondary antibody (1:200) for 1–2 h at 25 °C. The apoptosis of blastocysts was detected with a TUNEL kit (Beyotime, Shanghai, China). Hoechst 33342 (10 μg/mL in PBS) was used for DNA counterstaining. The fluorescence intensity of oocytes was observed using a fluorescence microscope (Olympus Tokyo, Japan). Oocytes were monitored at excitation/emission wavelengths of 488/530 nm and 371/464 nm.

### 4.6. Measurement of mtDNA Copy Number

The measurement of the mtDNA copy number was based on a previous study [45]. To extract DNA, 10 denude oocytes were collected in 2 µL of PBS. Oocytes were centrifuged at 12,000× *g* for 30 s at 4 °C. A volume of 8 µL of 5 mM Tris-HCl was added to each sample, and all samples were heated to 95 °C for 10 min. Proteinase K was added to the samples. Samples were heated at 55 °C for 30 min followed by heating at 95 °C for 10 min for proteinase K deactivation. The lysate was used for the measurement of the mtDNA copy number. The mtDNA copy number was measured using SYBR Green PCR Master Mix and a Rotor-GeneTM 6000 real-time rotary analyzer.

### 4.7. Real-Time Reverse Transcriptase-Polymerase Chain Reaction (RT-PCR) 

Total mRNA was extracted using Arcturus^®^ PicoPure^®^ RNA Isolation Kit (Thermo, KIT0204). For the specific methods, please refer to the kit manual. The concentration and purity of the extracted RNA were quantified using a NanoDrop 2000C spectrophotometer, and a PrimeScriptTM RT kit was used to synthesize cDNA. All the primers used in this experiment were synthesized by Sangon Bioengineering Co., Ltd. (Shanghai, China), and their sequences are shown in Table 1. The 18S RNA gene was used as a reference gene, and the standard curves of all the primers were verified. RT–PCR was performed using SYBR Green PCR Master Mix and a Rotor-GeneTM 6000 real-time rotary analyzer.

### 4.8. IVF and Embryo Culture 

After the control and TBBPA-treated groups were cultured for 22 h, the matured oocytes were washed and cultured in fertilization medium (50 μL), overlaid with mineral oil, and incubated in a humidified atmosphere of 5% CO_2_ at 38.5 °C. Frozen bull semen straws were thawed by immersing them in a water bath at 37.5 °C for 12 sec. Sperm was then gradient-centrifuged using 90% percoll solution 700 rcf for 15 min. The 90% percoll solution comprised 46.75 mg/mL NaCl, 2.3 mg/mL KCl, 0.35 mg/mL NaH_2_PO_4_, 23.8 mg/mL HEPES, and 2.09 mg/mL NaHCO_3_. Subsequently, sperm was centrifuged with TALP medium at 300 rcf for 5 min. The TALP medium comprised 5.84 mg/mL NaCl, 0.23 mg/mL KCl, 0.31 mg/mL CaCl_2_, 0.035 mg/mL NaH_2_PO_4_, 0.08 mg/mL MgCl_2_, and 2.1 mg/mL NaHCO_3_. The matured oocytes were co-incubated with spermatozoa in the fertilization medium for 16 h in a humidified atmosphere of 5% CO_2_ at 38.5 °C. Twenty hours post-IVF, remaining cumulus cells were removed by vortexing, and presumptive zygotes were cultured in synthetic oviduct fluid (SOF) supplemented with 5% FBS until blastocysts hatched from the zona pellucida (days 8–10). In vitro culture (IVC) was carried out under mineral oil at 38.5 °C in an atmosphere of 5% CO_2_ and 5% O_2_ in air with maximum humidity.

### 4.9. Statistical Analyses 

The data obtained from three replicates were reported as means ± standard errors (SEMs). Statistical analyses were performed with the one-way analysis of variance (ANOVA) and the independent-sample *t*-test using SPSS software, version 26.0 (SPSS, Chicago, IL, USA). Figures were generated using the GraphPad Prism software package (version 6.01; GraphPad, La Jolla, CA, USA). The average immunofluorescence intensity was measured and analyzed with Image J software. RT-PCR results were analyzed with the 2^−ΔΔ^Ct method. *p* < 0.05 was considered to be statistically significant.

## 5. Conclusions

In summary, we discovered that TBBPA exposure destroyed bovine oocyte maturation because of the abnormity of nuclear and cytoplasmic maturation. The study showed that TBBPA exposure induced mitochondrial dysfunction, which resulted in oxidative stress and early apoptosis. In addition, mitochondrial dysfunction led to insufficient energy supply to disrupt spindle assembly and chromosome alignment via regulating pyruvate metabolism mediated by PDK3. Therefore, this study indicated that the toxicity of TBBPA to oocytes gave rise to mitochondrial dysfunction inducing oxidative stress and early apoptosis by regulating PDK3. 

## Figures and Tables

**Figure 1 molecules-27-08111-f001:**
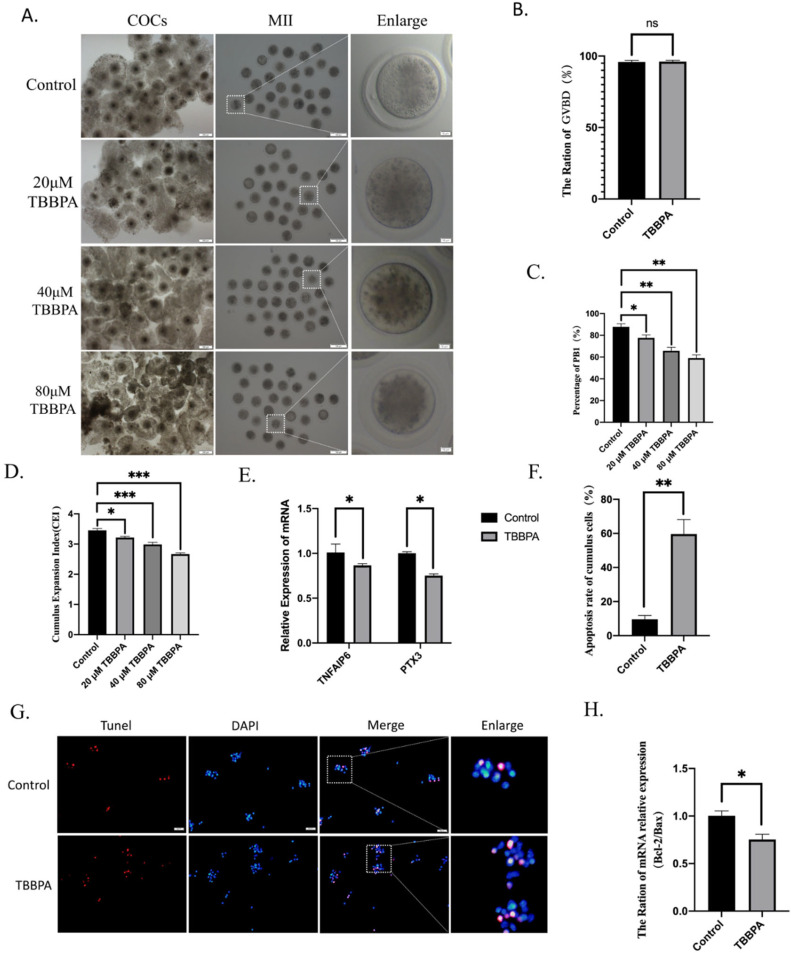
Effects of TBBPA exposure on bovine oocyte maturation: (**A**) Effects of TBBPA exposure on cumulus-cell expansion and polar-body extrusion in oocytes. COCs: scale bar = 200 μm. Metaphase II (MII) oocytes: scale bar = 100 μm. Enlarge: scale bar = 10 μm. (**B**) Ratio of GVBD in each group. Control, *n* = 144; TBBPA, *n* = 152. (**B**) Polar-body extrusion rate in each group. Control, *n* = 179; 20 μm TBBPA, *n* = 196; 40 μm TBBPA, *n* = 173; 80 μm TBBPA, *n* = 204. (**C**) The percentage of first polar-body extrusion was quantified in controls and groups with different concentrations of TBBPA. (**D**) Cumulus expansion index of each group. Control, *n* = 152; 20 μm TBBPA, *n* = 151; 40 μm TBBPA, *n* = 162; 80 μm TBBPA, *n* = 201. (**E)** mRNA expression levels of TNFAIP6 and PTX3. In total, 50 oocytes per group were used for RNA extraction. (**F**) Representative images of TUNEL staining of cumulus cells. Scale bar = 200 μm. (**G**) Apoptosis rate in each group. Ex/Em, 529/590; Ex/Em, 371/464 nm. (**H**) Ratio of BAX/BCL-2 in each group. * *p* < 0.05, ** *p* < 0.01, and *** *p* < 0.001, and ns *p* > 0.05.

**Figure 2 molecules-27-08111-f002:**
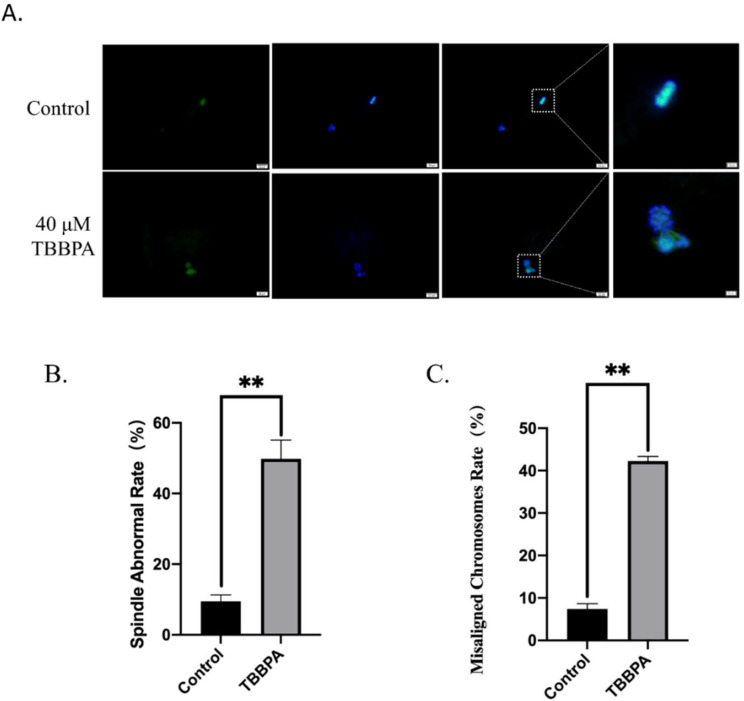
Effects of TBBPA on spindle/chromosome organization in oocytes: (**A**) Representative images of chromosome alignment and spindle morphology after TBBPA treatment. Ex/Em, 488/530 nm; Ex/Em, 371/464 nm. Scale bar = 20 μm. Enlarge: scale bar = 10 μm. (**B**) Spindle abnormal rate in each group. Control, *n* = 50; TBBPA, *n* = 45. (**C**) The percentage of misaligned chromosomes was quantified. Control, *n* = 50; TBBPA, *n* = 45. ** *p* < 0.01.

**Figure 3 molecules-27-08111-f003:**
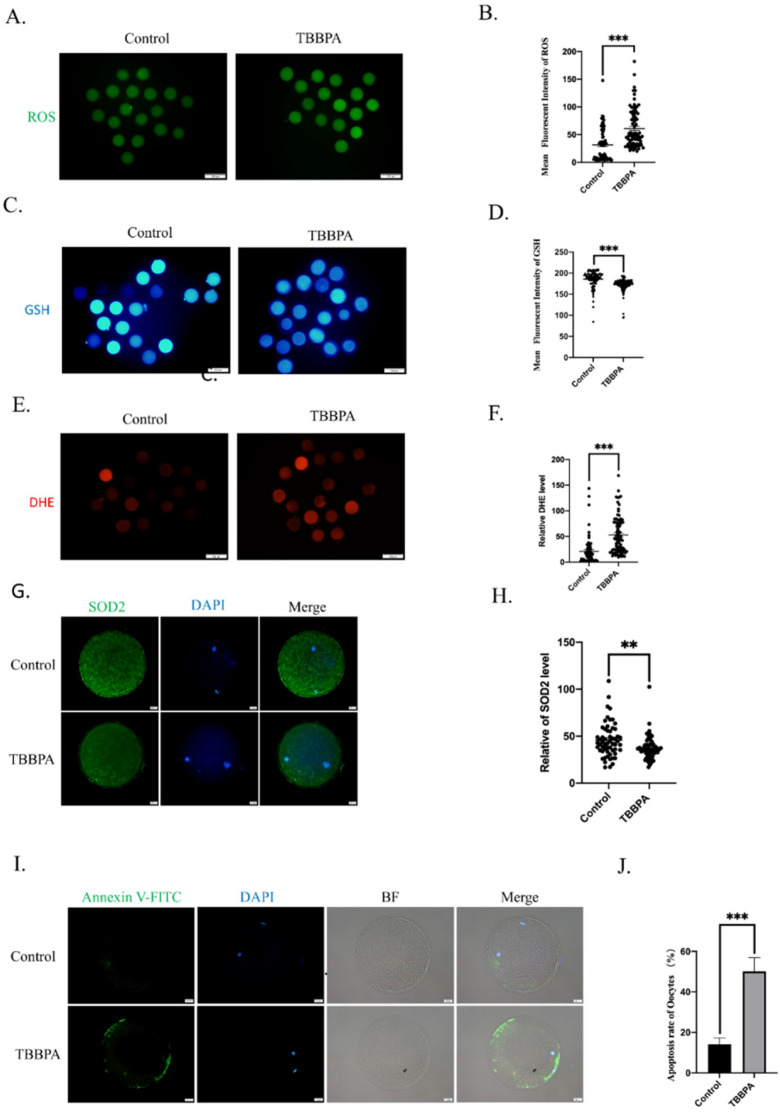
Effects of TBBPA on oxidative stress and apoptosis in oocytes: (**A**) Generation of ROS after TBBPA treatment. Ex/Em, 488/530 nm. Scale bar = 100 μm. (**B**) ROS levels were reflected by the mean fluorescent intensity of ROS. Control, *n =* 79; TBBPA, *n* = 81. (**C**) Representative images of GSH after TBBPA treatment. Ex/Em, 371/464 nm. Scale bar = 100 μm. (**D**) Mean fluorescent intensity of GSH. Control, *n* = 108; TBBPA, *n* = 114. (**E**) Representative images of DHE after TBBPA treatment. Ex/Em, 529/590. Scale bar = 100 μm. (**F**) Mean fluorescent intensity of DHE. Control, *n* = 74; TBBPA, *n* = 83. (**G**) Protein expression level of SOD2 after TBBPA treatment. Ex/Em, 488/530 nm. Scale bar = 100 μm. (**H**) Mean fluorescent intensity of SOD2. Control, *n* = 58; TBBPA, *n* = 56. (**I**) Annexin-V levels after TBBPA treatment. Ex/Em, 488/530 nm. Scale bar = 20 μm. (**J**) Apoptosis rate in each group. ** *p* < 0.01, *** *p* < 0.001.

**Figure 4 molecules-27-08111-f004:**
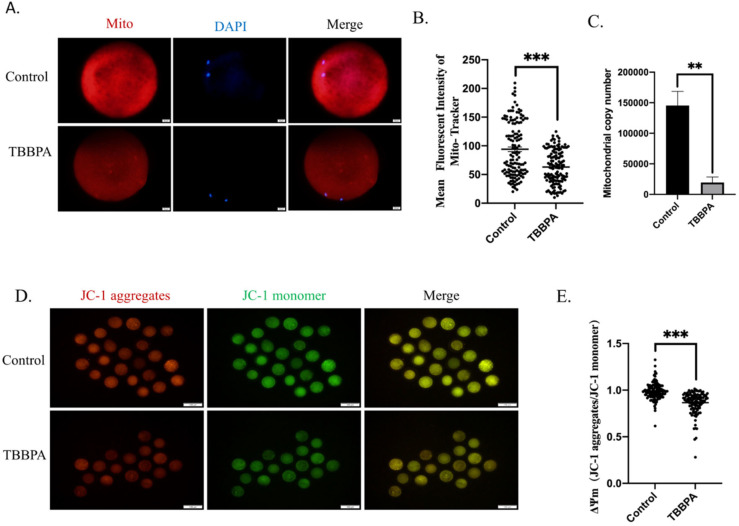
Effects of TBBPA on mitochondrial function in oocytes: (**A**) Representative images of mitochondria in each group. Ex/Em, 529/590. Scale bar = 20 μm. (**B**) Mean fluorescent intensity of Mito-Tracker. Control, *n* = 134; TBBPA, *n* = 138. (**C**) mtDNA copy number was measured. (**D**) Mitochondrial membrane potential after TBBPA treatment. Ex/Em,: 488/530 nm; Ex/Em, 529/590. Scale bar = 20 μm. (**E**) ΔΨm after TBBPA treatment (JC-1 aggregates/ JC-1 monomer). Control, *n* = 124; TBBPA, *n* = 116. ** *p* < 0.01, *** *p* < 0.001.

**Figure 5 molecules-27-08111-f005:**
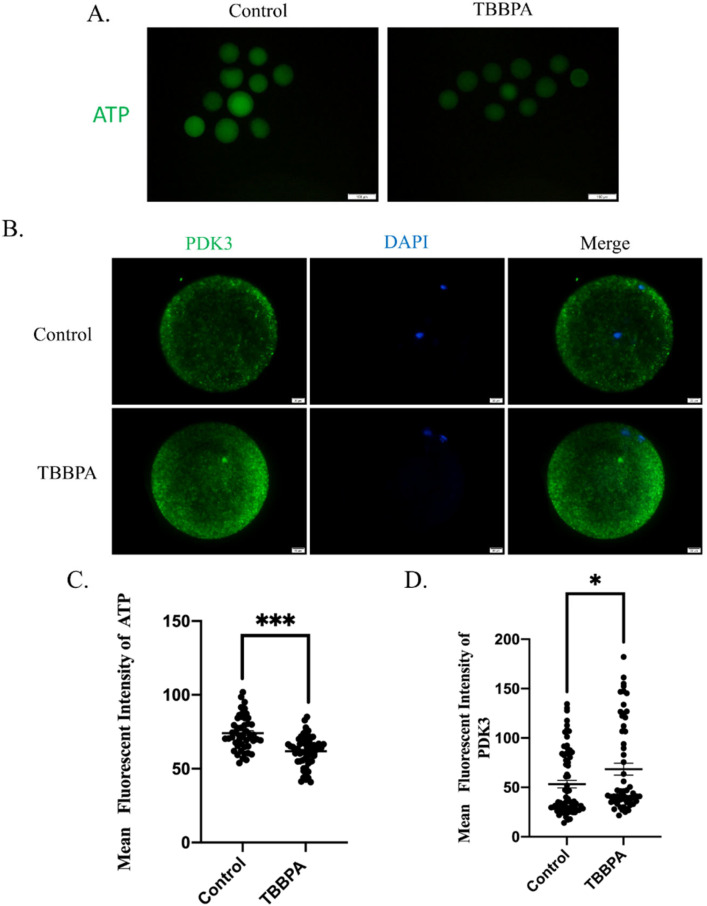
Effects of TBBPA on ATP levels in oocytes: (**A**) ATP levels after TBBPA treatment. Ex/Em, 488/530 nm. Scale bar = 100 μm. (**B**) PDK3 levels after TBBPA treatment. Ex/Em, 488/530 nm. Scale bar = 20 μm. (**C**) Mean fluorescent intensity of ATP. Control, *n* = 49; TBBPA, *n* = 58. (**D**) Mean fluorescent intensity of PDK3. Control, *n* = 73; TBBPA, *n* = 58. * *p* < 0.05 and *** *p* < 0.001.

**Figure 6 molecules-27-08111-f006:**
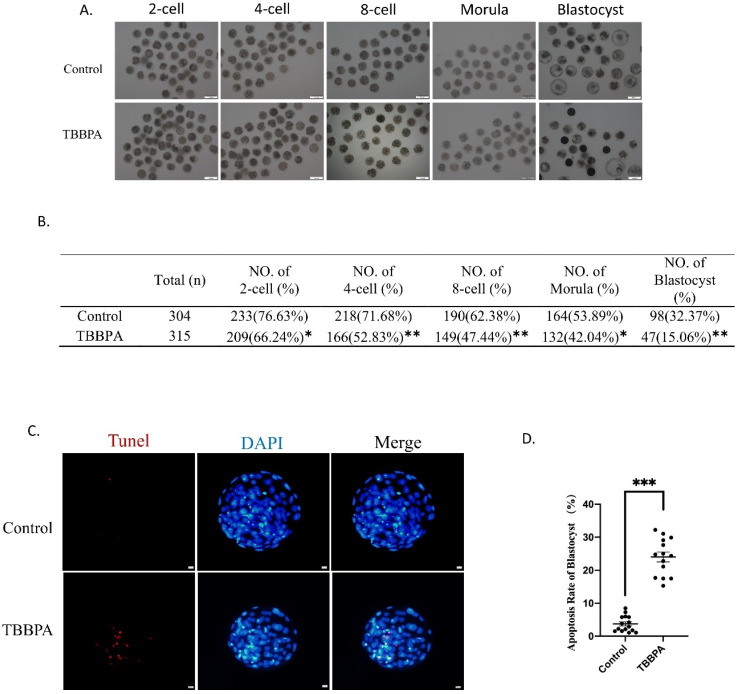
Effects of TBBPA on the developmental competence of oocytes: (**A**) Development of embryos in different stages after TBBPA treatment. Scale bar = 100 μm. (**B**) Developmental percentage of embryos in different stages. (**C**) Representative images of TUNEL staining of blastocysts in control and TBBPA groups. Ex/Em, 529/590; Ex/Em, 371/464 nm. Scale bar = 10 μm. (**D**) Apoptotic rate of blastocysts in each group. * *p* < 0.05, ** *p* < 0.01, and *** *p* < 0.001.

**Table 1 molecules-27-08111-t001:** Primers were used in this study.

Gene	Primer	Primer Sequence (5’–3’)
*18S*	Forward	GACTCATTGGCCCTGTAATTGGAATGAGTC
	Reverse	GCTGCTGGCACCAGACTTG
*Tnfaip6*	Forward	TATGGGAAGAGGCTCACGGATGG
	Reverse	GGTAGACGCCTGCTGCTTGTTC
*Ptx3*	Forward	TGGTCGCTGATGCTGTGATTTCC
	Reverse	GCCACCGAGTCACCATTTACCC
*Bcl-2 *	Forward	TCGTGGCCTTCTTTGAGTTCG
	Reverse	GCCTGTGGGCTTCACTTATGG
*Bax*	Forward	CGGAGATGAATTGGACAGTAAC
	Reverse	AGCACTCCAGCCACAAAGAT

## Data Availability

The data presented in this study are available upon request from the corresponding author.

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
