# Peer review of "Tetrabromobisphenol Exposure Impairs Bovine Oocyte Maturation by Inducing Mitochondrial Dysfunction"

_molecules, 2022, doi:10.3390/molecules27228111_

Round 1
Reviewer 1 Report
see file

Author Response
Point 1. Due to use in the study fluorescent probes, authors have to provide additional procedure details
(in Methods section, as well as indicate in the figures legends): specifically, indicate values of the emission and excitation wavelengths during measurement for each probe.
Response 1:Thanks for your suggestion. The manuscript was revised as follows:
Line 99-100: Ex/Em: 529/590; Ex/Em: 371/464 nm.
Line 118: Ex/Em: 488/530 nm; Ex/Em:371/464 nm.
Line 141: Ex/Em: 488/530 nm.
Line 142: Ex/Em:371/464 nm.
Line 144: Ex/Em:529/590.
Line 145: Ex/Em: 488/530 nm.
Line 146: Ex/Em: 488/530 nm.
Line 152: Ex/Em: 488/530 nm.
Line 166: Ex/Em:529/590.
Line 167-168: Ex/Em: 488/530 nm; Ex/Em:529/590.
Line 183-184: Ex/Em: 488/530 nm
Line 203: Ex/Em:529/590; Ex/Em:371/464 nm.
Line 308-314: For JC-1 detection, JC-1 monomeric form was observed with excitation/emission wavelengths of 488/530 nm, and the JC-1 aggregate form was observed with excitation/emission wavelengths of 529/590 nm. The oocytes were monitored with excitation/emission wavelengths of 488/530 nm to observe ROS, ATP, and apoptosis levels. GSH levels were observed with excitation/emission wavelengths of 371/464 nm. DHE and mitochondria were observed with excitation/emission wavelengths of 529/590 nm.
Line 326-327: The oocytes were monitored with excitation/emission wavelengths of 488/530 nm and 371/464 nm.
Point 2. Proofs are needed (reference or direct data) that TBBPA does not influence in any way fluorescent signals of the probes used.
Response 2:Thanks for your suggestion. The negative control was performed to conform probes specificity. Oocytes were treated with TBBPA, then images were captured with different wavelengths.
Point 3. Due to the nature of Mito-Tracker – its use in mitochondrial studies is limited either for direct detection of mitochondrial mass in the cell, or as an indicator of ΔΨm, but with understanding that it binds an accumulates only in mitochondria that possesses high potential. Therefore, authors shall clarify in the Methods and figure legend: do presented results reflect staining of mitochondria with Mito-Tracker in sets of separate experiments: 1) control cells + Mito-Tracker, and 2) incubation of separate group of cells with TBBPA with following staining with MitoTracker, or is this a set of continuous experiment: control cells + Mito-Tracker -> Signal measurement, then incubation cells with TBBPA -> Signal measurement?
Response 3:Thanks for your suggestion. In this study, COCs were randomly divided into different groups with or without TBBPA. After 22h culture, MII oocytes from different groups were collected for detection. Oocytes from different groups were treated separately with probes, then detected with the same condition. Control oocytes + Mito-Tracker, TBBPA treated oocytes + Mito-Tracker, then signal measurement. The manuscript was revised as follows:
Line 299-300: Denuded MII-stage oocytes from different groups were separate treated with fluores-cent probes. Fluorescent signals were captured with the same condition.
Point 4. How was experiment designed in case of the use of JC-1?
Response 4: Mitochondria is one of the most important organelles during oocyte maturation. Mitochondrial membrane potential (ΔΨm) is an important index to evaluate mitochondrial function and activity comprehensively. ΔΨm is a prerequisite for mitochondrial oxidative phosphorylation and the production of ATP. At the same time, it’s conducive to maintain normal physiological functions of cells. Therefore, JC-1 staining was applied in this study. Oocytes from different groups were collected to evaluate mitochondrial function. ΔΨm was detected by JC-1 staining in control and TBBPA groups.
Line 156-160: ΔΨm is a prerequisite for mitochondrial oxidative phosphorylation and the production of ATP. At the same time, it’s conducive to maintaining normal physiological functions of cells24. Oocytes from different groups were collected to evaluate mitochondrial function. ΔΨm was detected by JC-1 staining in control and TBBPA groups.
Point 5. I kind of confused with data presented in figure 6B and data in the text on p. 8… It is not clear to me: in zygote stage, number of cells for control and TBBPA-treated cells were the same? If yes,
then it is strange, and explanation will be needed; if not, then % of TBBPA-treated cells in zygote
stage shall be shown as an actual %, and so % for other stages of TBBPA-treated cells. Only after
that authors can recalculate and present data as relative to zygote stage taken as 100%.
Response 5: Thanks for your suggestion. The number of cells for control and TBBPA-treated cells were not the same. After 22h culture, all COCs were fertilized. It’s difficult to distinguish zygote by pronucleus for bovine. The development rate was defined as the ratio of embryos/ COCs. We revised the development rate as a form that displays the number of embryos at different stages.

Reviewer 2 Report
In this article, the effect and mechanism of TBBPA on oocyte maturation and development competence was determined, and the oocyte development competence was also studied. The research has important meaning for revealing impairing oocyte maturation and developmental upon TBBPA exposure. Before publication, several problems to be solved. My major comments are as follows.
1. In last paragraph of introduction section, it is necessary to further point out the innovation of this study.
2. It should be quoted relevant literatures in the Sentence “Previous studies demonstrated that the toxicity of TBBPA was associated with oxidative stress.”
3. In Line 127, please further explain the impact of the spindle defects and chromosome misalignment.
4. In Conclusions, please elaborate the research conclusion in more detail.
5. There are many spelling and grammar mistakes, please carefully check and modify.
Author Response
Point 1. In last paragraph of introduction section, it is necessary to further point out the innovation of this study.
Response 1: Thanks for your suggestion. The manuscript was revised as follows:
Line 63-65: As a worldwide flame retardant, TBBPA toxicity has been demonstrated in previous studies18-20. However, there are few studies about the effect of TBBPA on domestic animal reproduction.
Line 70-72: It’s the first time to detect the toxicity of TBBPA on female gamete development, and the underlying mechanisms regarding how TBBPA affects oocyte maturation.
Point 2. It should be quoted relevant literatures in the Sentence “Previous studies demonstrated that the toxicity of TBBPA was associated with oxidative stress.”
Response 2: We’re sorry for our carelessness. The manuscript was revised as follows:
Line 122-123: Previous studies demonstrated that the toxicity of TBBPA was associated with oxidative stress21-23
21 Suh, K. S. et al. Tetrabromobisphenol A induces cellular damages in pancreatic β-cells in vitro. J Environ Sci Health A Tox Hazard Subst Environ Eng 52, 624-631, doi:10.1080/10934529.2017.1294964 (2017).
22 Wang, J. & Dai, G. D. Comparative Effects of Brominated Flame Retardants BDE-209, TBBPA, and HBCD on Neurotoxicity in Mice. Chem Res Toxicol 35, 1512-1518, doi:10.1021/acs.chemrestox.2c00126 (2022).
23 Cho, J. H. et al. Tetrabromobisphenol A-Induced Apoptosis in Neural Stem Cells Through Oxidative Stress and Mitochondrial Dysfunction. Neurotox Res 38, 74-85, doi:10.1007/s12640-020-00179-z (2020).
Point 3. In Line 127, please further explain the impact of the spindle defects and chromosome misalignment.
Response 3: Thanks for your suggestion. The manuscript was revised as follows:
Line 109-114: Oocyte maturation is a precise-coordinated process that is controlled by the spindle assembly and chromosome alignment. An increased frequency of abnormal spindle and chromosome misalignment presented that the microtubule stability and attachment between spindle and chromosome may be impaired after TBBPA exposure. These abnormalities can induce fertilization failure and subsequent embryonic development arrest.
Point 4. In Conclusions, please elaborate the research conclusion in more detail.
Response 4:Thanks for your suggestion. The manuscript was revised as follows:
Line 375-382: In summary, we discovered that TBBPA exposure destroyed bovine oocyte maturation because of the abnormity of nuclear and cytoplasmic maturation. The study showed that TBBPA exposure induced mitochondrial dysfunction which resulted in oxidative stress and early apoptosis. In addition, the mitochondrial dysfunction led to insufficient energy supply to disrupt spindle assembly and chromosome alignment via regulating pyruvate metabolism mediated by PDK3. Therefore, this study implied the toxicity of TBBPA on oocytes gave rise to mitochondrial dysfunction to induce oxidative stress and early apoptosis via regulating PDK3.
Point 5. There are many spelling and grammar mistakes, please carefully check and modify.
Response 5: Thanks for your suggestion. We have checked spelling and grammar mistakes throughout the manuscript and revised these. The revisions have been marked in the manuscript.
